# Empanelment of the Population to the Primary Medical Care Institution of Sri Lanka: A Mixed-Methods Study on Outcomes and Challenges

**DOI:** 10.3390/healthcare11040575

**Published:** 2023-02-15

**Authors:** Pruthu Thekkur, Divya Nair, Manoj Fernando, Ajay M. V. Kumar, Srinath Satyanarayana, Nadeeka Chandraratne, Amila Chandrasiri, Deepika Eranjanie Attygalle, Hideki Higashi, Jayasundara Bandara, Selma Dar Berger, Anthony D. Harries

**Affiliations:** 1Centre for Operational Research, International Union Against Tuberculosis and Lung Disease (The Union), 75001 Paris, France; 2Department of Health Promotion, Rajarata University of Sri Lanka, Mihintale, Anuradhapura 50300, Sri Lanka; 3The Union-South East Asia (USEA) Office, New Delhi 110016, India; 4Yenepoya Medical College, Yenepoya (Deemed to be University), Mangalore 575018, India; 5The Foundation for Health Promotion, 21/1 Kahawita Road, Dehiwala 10350, Sri Lanka; 6Department of Community Medicine, Faculty of Medicine, University of Colombo, Colombo 00300, Sri Lanka; 7The World Bank, Colombo 00300, Sri Lanka; 8Project Management Unit, Primary Health Care System Strengthening Project (PSSP), Colombo 00300, Sri Lanka; 9Department of Clinical Research, Faculty of Infectious and Tropical Diseases, London School of Hygiene and Tropical Medicine, London WC1E 7HT, UK

**Keywords:** empanelment, rostering, digital health, perspectives, primary health care, health system, operational research

## Abstract

The registration of individuals with designated primary medical care institutions (PMCIs) is a key step towards their empanelment with these PMCIs, supported by the Primary Health Care System Strengthening Project in Sri Lanka. We conducted an explanatory mixed-methods study to assess the extent of registration at nine selected PMCIs and understand the challenges therein. By June 2021, 36,999 (19.2%, 95% CI-19.0–19.4%) of the 192,358 catchment population allotted to these PMCIs were registered. At this rate, only 50% coverage would be achieved by the end of the project (December 2023). Proportions of those aged <35 years and males among those registered were lower compared to their general population distribution. Awareness activities regarding registration were conducted in most of the PMCIs, but awareness in the community was low. Poor registration coverage was due to a lack of dedicated staff for registration, misconceptions of health care workers about individuals needing to be registered, reliance on opportunistic or passive registration, and lack of monitoring mechanisms; these were further compounded by the COVID-19 pandemic. Moving forward, there is an urgent need to address these challenges to improve registration coverage and ensure that all individuals are empaneled before the close of the project for it to have a meaningful impact.

## 1. Introduction

The Astana Declaration released during the Global Conference on Primary Health Care in 2018 reaffirmed that primary health care (PHC) remains a cornerstone of a sustainable health system for universal health coverage and health-related Sustainable Development Goals [1]. In 2019, the World Health Assembly adopted a resolution that recognized the role of primary health care in providing health services throughout a person’s life course, including prevention, treatment, rehabilitation and palliative care [2]. 

One of the foundational strategies for improving PHC, especially with regards to providing continuity of care, is empanelment [3]. Empanelment is a continuous, iterative set of processes that identify and assign populations to facilities, care teams or providers who in turn have a responsibility to know their assigned population and proactively deliver coordinated primary health care towards achieving universal health coverage [4]. It is also referred to as rostering, formal attachment, enrolment or patient registration in different care provision settings. This process helps in improving access of patients to healthcare; fosters relational, managerial, and informational continuity over time; smoothens referrals and coordination between multi-disciplinary teams; and enables better management of supply and demand, ultimately enhancing efficiency and quality of care [3,4]. 

Empanelment is becoming a key part of recent reforms in primary health care in different parts of the world, including lower-middle income countries (LMICs) [5,6]. Sri Lanka, a LMIC in the Southeast Asian region, has been going through a process of re-organisation of PHC under the World Bank supported Primary Health Care System Strengthening Project (PSSP) [7,8]. This project, launched in 2018, aims to increase the utilization and quality of PHC services, with an emphasis on detection and management of non-communicable diseases (NCD) in the country [8]. Empanelment of the population to a Primary Medical Care Institution (PMCI) was one of the first steps undertaken under this project towards re-organisation of PHC. In brief, this entailed (i) demarcation of geographic catchment areas for each PMCI and mapping of referral facilities with PMCI, (ii) registering each individual residing in the catchment area by assigning a unique personal health number (PHN), and (iii) ensuring that personal health records of these individuals are continuously updated at every encounter with the health system [8,9].

Though empanelment is considered to be a successful strategy for building PHC systems, there is relatively little information on the experiences around implementing empanelment in primary care settings from LMICs [10]. In Sri Lanka, as the first phase of PSSP, the empanelment process including registration was initiated in 63 PMCIs from June 2019 and the project plans to empanel the entire population in 550 PMCIs by December 2023. This first operational research study assessed the early experiences and challenges in empanelment and registration of the population to help guide future scale-up of empanelment in the country. Such a study requires both quantitative and qualitative methods for comprehensively understanding the outcomes of the activity and associated challenges [11,12]. Thus, we conducted a mixed-methods assessment of the empanelment process under the PSSP project in Sri Lanka to assess the extent of registration and understand the contextual challenges in implementing this activity. 

## 2. Materials and Methods

### 2.1. Study Design

This was an explanatory mixed-methods study [11]. The quantitative component was a longitudinal descriptive study involving secondary data extraction from the facility records and Health Management Information System (HMIS). This was followed by the qualitative component, which was a descriptive study involving in-depth interviews among healthcare workers and patients. 

### 2.2. Study Setting

#### 2.2.1. General Setting

Sri Lanka is an island country with a population of 21.8 million as of 2019 [13]. Administratively, the country is divided into nine provinces, each governed by an autonomous provincial council. The provinces are further divided into districts, administered by a district secretariat [14]. The most peripheral local administrative units are Grama Niladhari (GN) divisions [15].

#### 2.2.2. Specific Setting

In the Sri Lankan public health system, PMCIs are primarily responsible for provision of primary curative care. There are 986 PMCIs in the country and they are usually governed by the provincial health departments [16]. PMCIs include Primary medical care units (PMCU) and divisional hospitals (DH) [16]. While PMCUs provide only outpatient services, DHs provide outpatient, inpatient, and obstetric services [17]. Some PMCIs have a dedicated Healthy Lifestyle Centre (HLC) for providing preventive NCD care. Until recently, PMCIs did not have a specific catchment area. Non-specialist allopathic medical doctor and nursing staff provide services in PMCIs [18]. The PSSP focuses on reorganizing PMCIs to enable provision of patient-centered curative services, especially NCD care. 

Empanelment of the population is one of the preliminary activities undertaken in PMCIs supported by the PSSP. A list of people assigned to a given PMCI is referred to as a panel. The process of empanelment comprises three components: (1) identifying the panel for PMCI; (2) registration of the identified population; and (3) actively reviewing and updating panel data [8,9]. 

##### Identification of the Panel for PMCI

The PMCIs and the GN within the district are listed. The population of a given GN division is assigned to the closest PMCI based on travel times calculated using the geographic information system (GIS). All the individuals in a GN division are expected to be empanelled to a single PMCI. The process of identifying the panel for all the 986 PMCIs in the country has been completed centrally by the project management unit (PMU) and Ministry of Health (MoH). The average population for a panel per PMCI was 17,100 with 82% of the PMCIs having a population ranging between 3000 and 25,000 individuals [9]. After identifying the catchment areas for PMCIs, a secondary or tertiary institution is identified for each PMCI as a referral facility. The apex secondary or tertiary care institution and the group of PMCIs that link to this institution form a cluster (Figure 1). The staff in the PMCIs can reach out to the administrative staff of the assigned GN divisions and avail the master list of the population (either the electoral list or the list maintained by Grama Sevak/Niladhari) residing in the GN division to plan and carry out registration of the population of the identified panel [9].

##### Registration of Identified Population with Personal Health Number

The most important component of the empanelment process is the registration and issue of a PHN to each individual in the catchment population identified for the PMCI. The demographic details of the individuals (age, gender, and GN division) are entered into the online HMIS, which autogenerates a PHN. A PHN is a unique number assigned for each individual which can be used to link the individual with their health record. Those issued a PHN are considered to be successfully registered. Also, the individuals are expected to be educated on the importance of their PHN and the requirement of using this number at all interactions with the health system irrespective of level of care.

First, public awareness campaigns were conducted involving GNs, members of friends of facility committees (hospital management committee with community members for community engagement) and other community leaders, and the people were mobilized to the PMCIs for registration. People visiting the PMCIs were registered (passive registration). To improve the coverage, the PMCIs were encouraged to adopt opportunistic (registration of people visiting PMCI for treatment) registration, conduct outreach registration camps in the GN divisions, establish night/evening clinics for registration of the employed population, and also try active registration through house-to-house visits by healthcare workers (HCWs) or volunteers. The medical officer in-charge of the PMCI and staff nurse trained on the PSSP activities are primarily responsible for ensuring registration in the catchment area. Data entry operators (DEOs), if present in the facility, help with registration through the online HMIS.

##### Review and Updating the Panel with Personal Health Record

A Personal Health Record (PHR) is created for each person along with their PHN. The PHR is expected to be updated at every encounter with a health care provider. An electronic PHR is expected to be available in each facility once the national e-information system is fully implemented. The e-format will have information on services received by an individual and the different levels at which services have been provided. The database should be accessible to providers based on the need and the level of authorization.

Currently, the PHR is a paper record with the unique identifier (PHN) made available to patients at the time of registration. The PHR must have the date of the creation of record, date of updating the record for each subsequent facility visit, and personal details including PHN, name, date of birth, gender, contact phone numbers, and GN division of the current address. The morbidity details including body mass index, 10-year cardio-vascular disease (CVD) risk prediction, diagnosis according to International Classification of Primary Care (Second Edition), and treatment availed with a list of medications are to be mentioned. The patient is expected to bring the paper-based PHR during each health facility visit so that it is kept updated. 

### 2.3. Study Population

#### 2.3.1. Quantitative Component

We selected nine PMCIs out of the 63 PMCIs in Sri Lanka. To represent the 2:1 rural-urban population composition of the country, we used simple random sampling (computer generated numbers) to select six PMCIs from all PMCIs in rural areas and three PMCIs from all PMCIs in urban areas. The nine PMCIs also represent the nine provinces of the country. In each PMCI, we included all the individuals whose demographic details were documented electronically on HMIS and provided with their unique personal health numbers from June 2019 (implementation of PSSP) to mid-June 2021. These individuals were considered as successfully “registered”.

#### 2.3.2. Qualitative Component 

The medical officers, programme managers and the general public were interviewed to understand the process and challenges of empanelment. Purposive sampling using the ‘percentage of registration’ was used to select four PMCIs from where the participants were selected for an in-depth interview. One PMCI each with a high and low percentage of empanelment in urban and rural areas were included. Medical officers of these PMCIs and the programme managers of the district to which the PMCI belonged were interviewed. The general public residing in the catchment areas of selected PMCIs who had been successfully registered and those who had failed to register were interviewed, while attempting to ensure representation of gender and age groups. In total, we interviewed 28 people: medical officers (4), programme managers from MoH (4), nursing officers (3), a public health nurse (1), a data entry operator (1), individuals empaneled (8), and not empaneled (7).

### 2.4. Data Collection, Study Variables, Data Source and Study Tools

#### 2.4.1. Quantitative Component 

Trained research assistants used a structured proforma (Appendix A) to collect information from the medical officer and/or nursing staff regarding the GN divisions identified and the population assigned to the PMCI. The information on the various activities conducted to generate awareness about empanelment to create demand for registration and the approach used for registration in the PMCI were collected. 

The aggregate data on the total population (as per 2012 census of Sri Lanka) in the allotted PMCIs, stratified by assigned GN divisions and age (less than 35 years and ≥35 years), were obtained from the PMU of the PSSP. The number of individuals registered (stratified by GN divisions, age groups, and gender) was extracted from the electronic database of the registration module of HMIS. During July 2021, we extracted the data on registration from April 2019 to June 2021. 

Registration included collection and documentation of demographic details in the registration module of HMIS, auto-generation of the PHN through HMIS, screening for NCD risk (in those ≥35 years or increased risk of NCD) and issuing of paper-based PHR to the registered individual. We collected information from the medical officer/staff nurse on the extent of issuing PHR to those registered (not issued/partially issued/issued to all registered). 

#### 2.4.2. Qualitative Component

All the in-depth interviews were conducted by the research consultants who were medical doctors familiar with the health system in Sri Lanka, fluent in the local languages (Sinhala and Tamil), and trained and experienced in qualitative research. The interviews were conducted ten days after the checklist-based assessment and the findings of the quantitative study were used to amend the interview guide for the qualitative exploration. Separate sets of interview schedules (Appendix A) with probes were used for interviews among healthcare providers and patients. The interviews were audio recorded and were used to prepare the transcripts. On average, the duration of the interviews was 11.32 min (range 3.35 to 45.05 min). 

### 2.5. Data Analysis

#### 2.5.1. Quantitative Component 

The data extracted from the HMIS was entered directly into an EpiCollect5 application (Wellcome Sanger Institute, Cambridge, UK), a mobile phone-based data capture tool. Stata version 12 (StataCorp LP, College Station, TX, USA) was used for analysis. The frequency and percentage per nine selected PMCIs were used to summarize the details of type of registration and awareness-generation activities conducted by the PMCI. Using the aggregate data on total population from the census of 2012 and the ‘number of registered’ persons derived from the HMIS registration database, the below indicators were deduced to describe the extent of registration in the selected PMCIs and their GN divisions: The percentage of the total population registered in the PMCI: the numerator is all individuals registered until censor date and the denominator is total population in the PMCI area as per the 2012 census of Sri Lanka;The monthly trend in percentage of the total registered per PMCI from June 2019 to June 2021: The total percentage registered at the end of each month is calculated with the total number of individuals registered until the end of each month as the numerator and the total population in the PMCI area as per the 2012 census of Sri Lanka as the denominator. Assuming a similar trend (linear) we projected the percentage of the total that would be registered by December 2021, December 2022, and December 2023 (end of project). For this analysis only eight PMCIs which initiated registration since June 2019 were included;The percentage of the individuals aged ≥35 years registered in the PMCI: The numerator is individuals aged ≥35 years registered until censor date and the denominator is the total number of individuals aged ≥35 years in the PMCI area as per the 2012 census of Sri Lanka. The age cut off of 35 years was chosen as the PSSP had a mandate for screening individuals aged ≥35 years for NCD risk factors;The percentage of males among all the individuals registered in the PMCI: the percentage was calculated with the total number of males registered in the PMCI as the numerator and the total number of individuals registered in the PMCI as the denominator;The median (IQR) percentage of the total population registered in the GN divisions of the selected PMCIs: The percentage was calculated with individuals registered from each GN division as numerator and the total population in the respective GN division as per the 2012 census of Sri Lanka as denominator. The median (IQR) of the percentages calculated for each GN division was deduced.

#### 2.5.2. Qualitative Component

The transcripts were prepared on the same day of the interview with the use of notes and audio recording. Thematic analysis was done by the PI (PTK) using Atlas-Ti software to identify themes on the challenges in the strengthening and reorganisation of PMCIs. The second investigator (AMVK) reviewed the analysis and decision on coding/categories and theme generation was done in consensus. Similar codes were combined into categories to describe certain themes. To ensure that the results reflected the data, the codes/categories were related back to the original data. The findings were reported as per ‘Consolidated Criteria for Reporting Qualitative Research’ guidelines [19]. 

## 3. Results

### 3.1. Quantitative Component

#### 3.1.1. Identification of Panel for PMCI

In all the nine (100%) selected PMCIs, the GN divisions had been identified (assigned) and the list of the assigned GN divisions had been communicated to the PMCI. In total, 150 GN divisions were assigned to the nine selected PMCIs. The median (range) number of GN divisions assigned per PMCI was 14 (2–30). The median (range) population assigned per PMCI was 15,607 (4000–55,033). The updated master list (as of 2019) with the line list of individuals in the assigned GN divisions and their demographic details (age, gender, and address) had been obtained from Grama Sevak/Niladhari in three PMCIs. 

For all the nine PMCIs, the secondary and/or tertiary hospitals for referral of the patients had been identified. In three PMCIs, the identified secondary and/or tertiary hospital were outside the province where the PMCI was located.

#### 3.1.2. Registration at the PMCI

##### Awareness Generation and Registration Process

All the PMCIs had conducted one or more awareness-generation activities in the assigned GN divisions to promote registration: eight PMCIs made mass announcements, seven displayed posters with information related to registration in public places, four reached out to individual households through pamphlets and/or phone calls, and in three the individuals visiting the hospital were informed about mobilising their neighbours to get registered in the PMCI. 

In all the nine PMCIs, those visiting the PMCI for registration (passive registration) and those visiting the PMCI for medical reasons (opportunistic registration) were registered. In seven PMCIs, outreach camps for NCD screening or registration were conducted in the GN divisions. One PMCI conducted ‘night clinics’ and two conducted ‘weekend clinics’. Only in one PMCI was active registration done through house-visits by the incentivized volunteers. 

##### Coverage of Registration and Estimated Coverage by December 2023

Of the total population of 192,538 (as per the 2012 census of Sri Lanka) assigned to the nine selected PMCIs, 36,999 (19.2%, 95% CI-19.0–19.4%) had been registered as of June 2021. The percentage of the total assigned population registered in each PMCI ranged from 2% (in a PMCI which started registration from March 2020) to 58%. 

The monthly trend in the progress of registration in eight PMCIs that implemented PSSP from June 2019 is shown in Figure 2. There was a plateau in the percentage registered during 2020 (COVID-19 pandemic period). Using the existing linear trend, we estimated the percentage of the total population that would be registered by the end of the year 2021, 2022, and 2023. By December 2021, two PMCIs would register >50% of their population. By December 2022, four PMCIs would register >50% of their population. By December 2023 (end of project), all the individuals would have been registered in only one PMCI. The aggregate percentage of the total population registered by December 2023 (end of project) was estimated to be 48.2%.

##### Coverage of Registration among Individuals Aged ≥35 Years

Of the 86,615 (as per the 2012 census of Sri Lanka) individuals aged ≥35 years assigned to the nine selected PMCIs, 31,299 (36.0%) had been registered. Thus, this age group comprised 84.6% of all the 36,999 individuals registered at nine selected PMCIs.

##### Gender Distribution of Registered Individuals

Of the 36,999 individuals registered in nine PMCIs, 14,504 (39.2%, 95% CI: 38.7–39.7%) were males. The median (range) percentage of males among the individuals registered in the nine PMCIs was 36.0% (28.8–50.4%). In six out of nine PMCIs the percentage of males registered was less than 40%, compared to 48% males among the adult population in Sri Lanka (2012 census). 

##### Coverage of Registration across the GN Division

At least one individual from the assigned GN division had been registered in the respective PMCI in 145 (96.7%) of the 150 GN divisions under the nine PMCIs. In 80 (53.3%) GN divisions, the percentage of the total population registered was ≤10% (Figure 3). The median (IQR) percentage of the population registered per GN division in 150 GN divisions was 9.1% (2.4–26.6%).

#### 3.1.3. Updating of Panel with Personal Health Records

##### Issuance and Completeness of Paper-Based PHR

There was no documentation of the number of paper-based PHRs issued in each PMCI. Therefore, we were not able to estimate the percentage of the registered individuals who had received PHR. The staff in charge of empanelment mentioned that in each PMCI, the issue of paper-based PHRs was pending for some of the individuals registered.

##### Availability of HMIS for Updating Electronic PHR

The HMIS module for updating the PHRs was installed in the computers and laptops available in all the selected PMCIs. However, only two PMCIs had initiated the process of updating the electronic PHRs during the visit of registered individuals to the medical clinic for care. 

### 3.2. Qualitative Component

In this section, we summarize the insights obtained from qualitative exploration regarding the challenges faced by HCWs in carrying out the registration process. Around 51 codes related to challenges were identified from the transcripts of the interviews among healthcare providers and the general public. The codes were grouped into categories under the three sub-themes: recipient-related, system-related, and provider-related challenges (Figure 4).

#### 3.2.1. Recipient Related

The four major challenges related to the general public who were the recipients of the registration process which could have had an implication on the coverage of the registration are summarised below. 

##### Lack of Awareness

The individuals who were not empaneled expressed that they were not aware of the registration conducted at the PMCIs, in spite of awareness campaigns on PSSP. A female who had not registered, when told about the registration and utility of it, said:
“No, I have never heard of that. I think such a system is not there in XXX. This is because XXX is a divisional hospital. I think the hospital is not developed to that level. Actually, they don’t use any computers. All registrations are done manually.”

##### Lack of Perceived Need for Registration

The HCWs felt that the general public did not turn up for registration despite reaching out to them through the awareness campaigns. A medical officer said,
“Sometimes we deliver the message to the community. But if we make 100 people aware, only 25 come. It’s very difficult to gather people. When we give a date for registration, they don’t come on that date…”.

The HCWs were of the opinion that the individuals who belonged to higher socio-economic strata and were educated did not consider registering as they usually visited the private health care facilities. A medical officer putting out his concern said,
“But one thing I noticed is the rich and educated people do not participate in this but the other poor villagers came and engaged in this… Because the rich go to private clinics.”

##### Lack of Trust on Identified PMCI

The HCWs and the general public felt that there is hesitancy among individuals to visit PMCIs due to a lack of trust. This lack of trust on PMCI is largely because of non-availability of all the healthcare services (laboratory investigations, specialist consultation, and drugs). One individual in the community, said,
“Almost all the people are going to XXX hospital (secondary hospital) for treatment. Because, here in YYY (PMCI) they have few facilities and there are also no specialist doctors. For every illness they are visiting XXX. But only for OPD some are visiting YYY. That is because of lack of facilities here. Even when someone visits for an emergency at night they are sending that person to YYY, because they don’t have enough facilities.”

##### Poor Access to PMCI

The HCWs feel that, though the GN divisions were assigned to the PMCI based on the travel time, the availability of transport determined the utilization of services in the assigned PMCI, including registration. A medical officer explained this by saying,
“Though individuals are empaneled to this PMCI, they do not utilize services from the hospital. When we consider XXX (GN division), people living along the main road (highway connecting two cities) tend to go to YYY (secondary hospital). They never come to (the assigned) PMCI as it is not convenient for them to travel to this PMCI, which is located far away from highways and there are no proper transport facilities.”

Also, individuals who were employed were not able to visit the PMCI for registration during working hours. A male who had not registered said,
“They (individuals in his village) went. They informed me too. But I was at work on that day. A very big crowd went on that day for the camp (registration).”

#### 3.2.2. System Related

Under the system related challenges, we have summarised the issues concerned relating to the systems put in place for registration. We have described six major challenges under this section.

##### Non-Availability of Population Details

The HCWs expressed that though the GN divisions were identified, there was not much support from the project to obtain a recent population database (master list). The HCWs of the PMCI had to contact several administrative officers to access a recent updated master list. This hindered micro-planning for registration-related activities. A medical officer while sharing his experience on the same issue, said,
“We don’t have details. We didn’t even receive voter’s registries. I requested that from Divisional Secretary, XXX. But he said that he is not authorized to give that and needs to ask from the YYY (District Secretary office). Finally, he said that he can’t give it.”

##### Challenges with HMIS and Generating PHN

The challenges related to the HMIS have been discussed in detail in our previous publication. Here, we describe some of the challenges with the online HMIS which had direct implications for generating PHN. As PHN could not be readily generated during outreach camps due to poor internet connectivity, the PHN had to be generated through other means. Using dummy data entered in to the HMIS, the PHN was generated prior to the camp and issued to the individuals attending the camp. Generating the PHN prior to the camp and manually writing it on paper led to transcription errors in the PHN. When the HCWs tried to update the demographic details against a PHN in the HMIS after the camp, often the PHNs would not match.

Alternatively, the PHN was generated and individuals were registered by transcribing the demographic details available in the paper-based ‘participant register’ (used during outreach camp). Generating PHN after the camp using data in the ‘participant register’ made it difficult to provide the PHN to the recipients. Though individuals were registered officially with PHN after the camp, the individuals remained unaware of their PHN. During informal discussions with the HCWs, it was found that in one of the PMCIs the generated PHNs were never given to the respective individuals. A nursing officer in charge of registration explaining the same said,
“The huge challenge we have is HMIS due to internet connection. Even in hospital, we are working with our personal hot spot. We are not able to readily generate PHN for registration. So, we need to generate PHN before camp or register the individual after camp. Sometimes because of this, we failed to give the PHN numbers to those registered.” 

##### Registration of Same Person Multiple Times

The HCWs felt that with the current system (HMIS), it was difficult to identify the individuals registered if the individuals fail to get their PHR (in some cases it was not issued in the initial days). Though some more unique identifiers such as national ID and the mobile numbers were entered into the HMIS, the system failed to pick up the duplicate entries. This led to registration of the same individual multiple times, leading to errors in estimation of targets and also a waste of resources. A medical officer explaining the issue with multiple registration of the same person, said,
“Actually, there are instances where we register a single patient 10–12 times. This is a waste of resources. We waste paper, money, and time.”

##### Lack of Transportation Support for Outreach Camps

The HCWs mentioned that there was a lack of transportation support for outreach camps. The ambulances at PMCIs were usually used for responding to emergencies and were not available for outreach camps. A medical officer highlighting this said,
“The problem is that when we want to have mobile clinics, the staff are lacking and transport is not available. Also, we can’t use the ambulance as well. Sometimes staff paid their own money for traveling to camp sites.”

##### Lack of Monitoring Mechanisms

During the interviews it was observed that there was very little internal and external monitoring of the process and progress of registration. The responsibility of registration was handed over to one or two nursing officers in the facility and largely went unsupervised. A medical officer narrated the story of a nursing officer who handled the registration activity for about five months but had registered only seven individuals on HMIS. He said,
“She (nursing officer) stated that data were entered (registration). But when we checked only 7 (people) were entered. So, we had to start from the beginning… There were no data. So, we started from the beginning…”

##### Challenges due to COVID and Its Response

The HCWs felt that the COVID-19 pandemic and its response led to a decline in registration. According to them, this was mainly because of the inability to conduct outreach camps and a reduced footfall in OPDs. The HCWs mentioned this as one of the major challenges they faced in improving the coverage of registration. A medical officer explaining the same said,
“The registration numbers collapsed with the onset of COVID-19. That count drastically dropped down as individuals were registered only when they came to OPD. Even OPD numbers went down with COVID-19 as our facility was functioning as a dedicated COVID hospital…”

#### 3.2.3. Provider Related

In this section we have described four challenges related to existing HCWs, which contributed to a low rate of registration in the PMCI. 

##### Lack of Dedicated Staff for Registration

The HCWs complained about the lack of dedicated staff especially public health nurses and data entry operators for carrying out registration and screening activities. According to HCWs, it was difficult to allot the existing staff (nursing officers and minor staff) for registration activities as they were required for healthcare service delivery at PMCI. A medical officer explaining the situation said,
“After the project started we registered patients daily from the outpatient department (OPD). But we didn’t have enough resources and time. The main problem was lack of staff. We had problems with assigning a dedicated nurse or other staff for this task.”

A nursing officer highlighting the issue of not having a dedicated staff for PSSP activity said,
“No, we didn’t have any opportunity (outreach activity). Basically, we didn’t have a dedicated health worker to organize this. We really need someone dedicated.”

##### Focus on Screening Targets over Registration

There were no targets set for registration of all the individuals (total population) in the identified area. As there was a disbursement-linked indicator related to coverage of screening among those aged ≥35 years (achievement of 50% by end of year 2), the medical officers and programme managers prioritized meeting the targets set for screening. Additionally, the HCWs assumed that the focus of the PSSP was only those aged ≥35 years, even for registration. A medical officer likened their achievement in screening to achievements in registration, saying,
“The population assigned to us was around 64,200, the eligible population (≥35 years) for registration was about 34,000. We had screened about 23,000 from 2017 itself. Thus, we were well ahead of the target for registration.”

On similar lines, the programme manager detailing the targets for registration said,
“In PSSP, we were given targets such as 25% registration per year. Assume, 40% of the total population were above 35 years in a particular catchment area, 25% had to be registered in year 1 and 50% by year 2.”

##### Confusion Regarding GN Divisions and Individuals to Be Registered

During the interviews, it was evident that some of the HCWs involved in the empanelment process were not aware of the GN divisions identified for their PMCI. They were confused between the Medical Officer of Health (MOH) area (the area demarcated for the preventive services of maternal and child health care) and that identified for PMCIs. A medical officer wrongly mentioning the number of GN divisions said,
“Hmmm (looks confused), I can’t remember exactly. As I remember it is around 20.”

A nursing officer of the same PMCI, said,
“I can’t remember exactly. As I remember it is 48 GN divisions (same as the MOH area).”

The HCWs were confused about who were eligible for the registration. Even during the awareness campaign, the emphasis was on registering only those aged ≥35 years. A 34-year male who had not registered, said,
“I didn’t go for registration. Last year when they were doing it (registration), it was definitely for older people. I’m not quite sure whether it was 35 or 40 (age). Not for us… I think even the banners said so (we verified by looking at the photographs, both the notice and the letter sent to households mentioned that the programme of registration was intended for people above 35 years).”

When asked about the reason for registering only those aged ≥35 years, HCWs mentioned that they received guidance from the Regional Director of Health Services (RDHS) to register (empanel) only those aged ≥35 years. A medical officer during the interview said,
“RDHS informed us to register all those aged 35 years and above in the given empanelment area.”

##### Negative Attitude and Rude Behaviour

The HCWs expressed that some of their colleagues had a negative impression about the PSSP (especially registration and screening) considering the amount of additional work it brought in. The HCWs of well-performing PMCIs felt that the negative attitude among the medical officers could be an important reason for deficiencies in implementation at other PMCIs. A medical officer talking on the same issue said,
“Some of the medical officers in charge of the PMCIs were very negative towards it. They didn’t want to take it over and increase their burden of work. In my opinion, I feel if you see any deficiencies in other institutions that would be the main reason. For example, the two experienced doctors at XXX and YYY got themselves transferred just because their PMCI was selected for implementing PSSP, and they didn’t want to exert themselves with this project.”

#### 3.2.4. Positive Aspect of Empanelment

The individuals who have had registered and received the PHRs appreciated the utility of the paper-based PHRs and also electronic PHR. The individuals felt that everyone in the country should receive the PHR as it can improve the quality of care. 

A female appreciating the utility of registration said,
“As you say if I get sick at a relative’s residence and they too are unaware of health conditions and when I am taken to hospital all my details will be available in the computer or book. That will be easy for everyone.”

Even those who had not registered, when told about the registration process and its utility during the interview, appreciated the potential utility of registration and PHN. One of the individuals who had not registered said,
“As one can access the data base (electronic PHR) via this number (PHN) it is good. I don’t know how far Sri Lanka uses this technology yet. But if the health system has this kind of thing it is valuable for quality care provision at hospitals.”

## 4. Discussion

This is the first study which comprehensively assessed the extent of registration for empanelment and challenges associated with it in nine PMCIs supported by PSSP since mid-2019. It is noteworthy that all the nine PMCIs had received the list of GN divisions identified for them and had initiated the registration process. Awareness-generation programmes about PSSP and the importance of registration were conducted by all the PMCIs. By June 2021, about 37,000 individuals were registered and issued a PHN which could be used for accessing their PHRs in the hospitals. The individuals registered appreciated the registration process and the potential for the continuum of care with the newly introduced system.

However, only 19% of the total population in the nine selected PMCIs were registered by June 2021. Going by the current pace of registration, only about 50% of the population would be registered by the end of the project (December 2023). Among those registered, there was an under-representation of the male gender and the age group < 35 years. In spite of the awareness activities conducted in most of the PMCIs, the community was not fully aware of the registration activity in the PMCIs. Major healthcare provider and system-related reasons for poor registration coverage were lack of dedicated healthcare staff for registration, misconceptions of HCWs about individuals to be registered, challenges with HMIS in generating PHN, non-availability of population details for planning, restricting to only opportunistic or passive registration, and lack of monitoring mechanisms. Registration and screening activities also slowed down drastically since March 2020 due to COVID-19.

This study had certain strengths. First, the findings from this study reflect the real-world situation of registration for empanelment as it was conducted within the routine programmatic setting of PMCIs. Second, we derived the percentage of individuals registered using the registration module of the HMIS database by considering only those individuals with PHN as registered. Therefore, these estimates of registration coverage are more objective and reliable compared to estimates derived using the ‘number of registrations in a quarter’ reported by the HCWs from the paper-based registers maintained in the PMCIs. Third, at least one PMCI was selected from each province of Sri Lanka, which improved representativeness of information, and we had a reasonable sample size for precisely estimating the percentage of individuals registered. Finally, qualitative interviews with HCWs, programme managers, and the general public provided in-depth insights into the reasons for the gaps in registration.

There are a few limitations to be considered when interpreting the results. First, we used census data from 2012 as the denominator for estimating the registration coverage in 2021. As the population would have increased due to the positive annual growth rate in Sri Lanka, we might have overestimated the percentage of individuals registered. Second, we used a linear forward projection for estimating the coverage by December 2023. However, the projection needs to be interpreted with caution due to small numbers of data points and also lack of a clear linear trend. Third, the total number by gender in each of the PMCI as per the 2012 census was not available. So, to comment on the participation and registration coverage among males we had to calculate the percentage of males among all those registered and compare it with the gender composition at the national level. Fourth, we did not evaluate the process of identification of panel (GN divisions) for PMCIs as this activity had been carried out centrally prior to the launch of PSSP at the PMCIs. The scope of the current study was limited to evaluation of the coverage and challenges associated with registration of this pre-identified panel. A detailed description of the process of panel identification have been provided in the Guidelines for Operationalizing Primary Medical Care Services in Sri Lanka [9].

The study provides insights on the potential challenges that any low- and-middle income country (LMIC) may face in empaneling and registering the general public for strengthening healthcare service provision [3]. This is relevant as some countries in the region have initiated similar measures as a step towards universal health coverage [20,21]. Based on the study findings, we put forward some recommendations which the PSSP may consider for improving the registration and, consequently, empanelment of the population.

First, the registration coverage has been low (19%) and only about 50% of the population would be registered by the end of the project (December 2023). Though awareness-generation activities were conducted, a few individuals mentioned that they were not aware of the registration activity and therefore did not register. To mitigate this, the staff of PMCIs should actively engage with their friends of facility committees and Grama Niladharies for planning and executing awareness-generation activities. As a mix of multi-media for awareness generation on health issues has been proven to be beneficial in LMICs, there is a need to conduct various media campaigns to promote registration by highlighting the importance of the PHN and PHRs [22].

Second, one of the major reasons for the low coverage of registration was the restriction to facility-based registration, either through opportunistic or passive registration. However, studies from Brazil, Costa Rica, and South Africa, which employed an active approach through house visits by HCWs, have reported better registration coverage and better engagement of the community [23,24,25]. Even in the current study, the facility which adopted active registration through house visits had better coverage (>50%) compared to other PMCIs. Thus, the project could consider house-to-house visits by incentivized community volunteers or public health nurse officers recruited for community activities to enumerate the individuals in each house, and generate and issue the PHN. It may not be possible to involve the existing staff for active registration, as they are preoccupied with healthcare service provision in the facility and might not be able to dedicate time for registration activities in the community [26]. Further, to plan and execute active registration efficiently, the PMCIs should obtain an updated master line list (electoral list or enumeration list) of individuals in each GN division from the Grama Sevaks. The PMU of PSSP through the Directorate of Health Services should send an official communication to ‘divisional secretariat office’ (revenue administrative office) requesting Grama Sevaks to support and share the master list with PMCIs for registration.

Third, though the PSSP intended to reach more individuals for registration through outreach clinics, the PMCIs were not able to conduct clinics regularly due to lack of dedicated staff and transportation facilities. Similar challenges in conducting outreach clinics have been reported from LMICs [27,28,29]. The conduct of outreach clinics was completely called off during COVID-19 pandemic period, as the COVID-19 response was prioritized over routine activities [30,31]. Now that COVID-19 cases have reduced, the PMCIs need to resume outreach clinics in each of the identified GN divisions for registration. Allocating a dedicated day for outreach clinics in each month for a GN division would improve the participation of individuals and ensure the smooth management of logistics (human resource and transport).

Fourth, the HCWs prioritized screening of individuals aged ≥35 years for NCD risk as the screening coverage was one of the disbursement-linked indicators of the project and closely monitored by the programme managers. There was a misconception among HCWs that only those eligible for screening needed to be registered. This led to registration of only those individuals aged ≥35 years (86% of total registered). As the registration and screening for NCD risk are two different activities under the project, there is need to re-orient HCWs and bring back the focus on registration to all the individuals of all ages.

Fifth, males constituted a relatively smaller portion (39%) of the registered population compared to the male gender distribution (48%) in the general population. This was largely because the men were mostly away at work and could not attend the registration clinics or camps whose timings usually coincided with their working hours. Anticipating this, the PSSP had recommended night clinics and weekend clinics for reaching out to the working class [8,9]. However, these were rarely conducted. Thus, efforts must be made to improve male participation in registration through targeted mobilization and conducting night clinics and weekend clinics. An indicator of ‘the cumulative proportion of males among the total registered’ can also be added to the list of indicators used to monitor registration.

Finally, there is a need for setting annual targets for the registration at GN divisions and PMCIs. The programme managers need to monitor these targets on a monthly or quarterly basis and focus on the GN divisions and PMCIs with poor registration coverage. The recommendations made in our previous publications to overcome staff shortages and resolve the issues with HMIS should also be taken into consideration to enhance efficiency of the registration process [26,32]. 

## 5. Conclusions

This first assessment of the extent of registration for empanelment in PMCIs showed that less than one-fifth were registered since the implementation of the PSSP in June 2019, and estimated that 50% of the population would be registered by the end of the project (December 2023). In addition, there was an under-representation of the male gender and the age group <35 years. The reasons for poor registration coverage included lack of dedicated healthcare staff for registration, misconceptions of HCWs about individuals to be registered, restricting to only opportunistic or passive registration and lack of monitoring mechanisms, which were further compounded by the COVID-19 pandemic situation. Moving forward, there is an urgent need to address these challenges to improve registration coverage and ensure all of the individuals are registered (empaneled) before the close of the project for it to have a meaningful impact.

## Figures and Tables

**Figure 1 healthcare-11-00575-f001:**
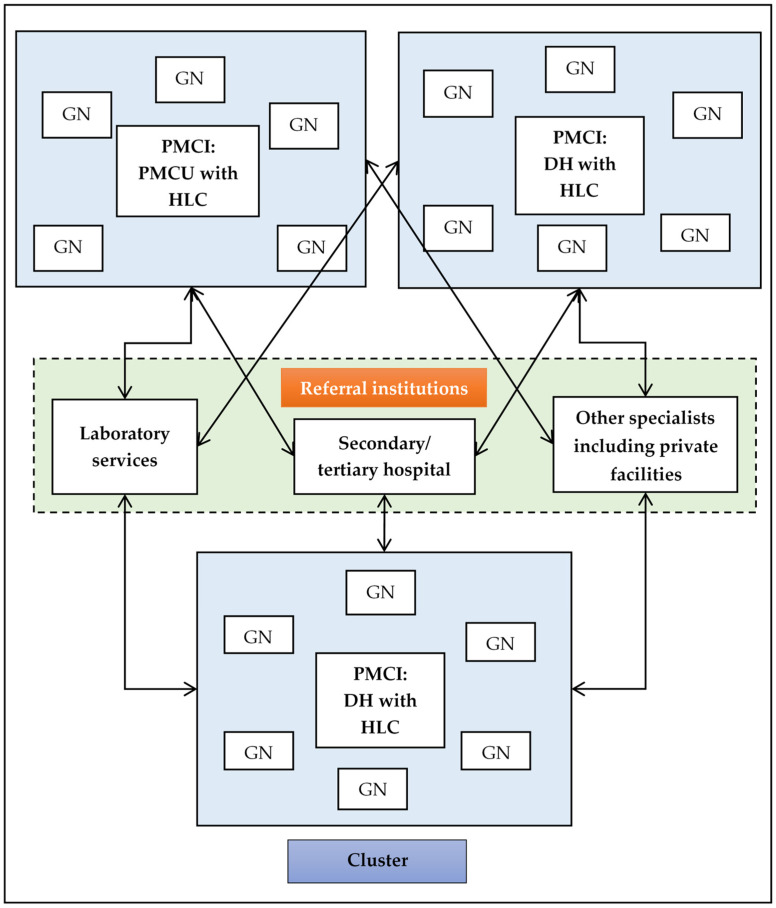
Schematic diagram representing identification of PMCI and cluster under the PSSP project. Abbreviation: DH = Division Hospital; DH-C: Division Hospital of Type-C; PMCI = Primary Medical Care Institution; GN = Grama Niladhari; PMCU = Primary Medical Care Unit; HLC = Health Lifestyle Centre.

**Figure 2 healthcare-11-00575-f002:**
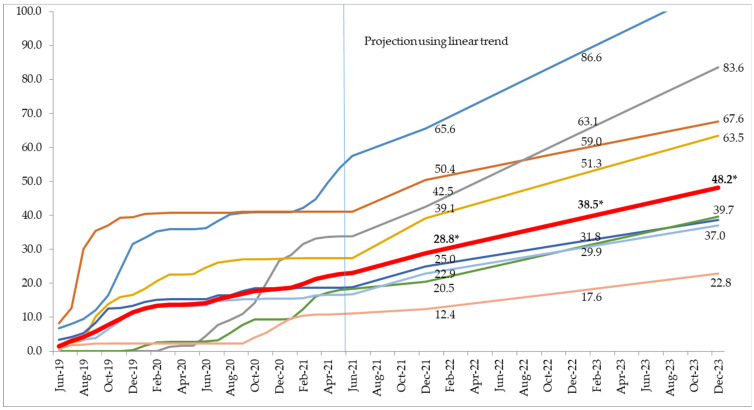
The monthly trend in the cumulative percentage of the total population registered during June 2019 to June 2021 and the linear forecast for December 2021, 2022, and 2023 in the eight selected PMCIs supported by the PSSP in Sri Lanka. Note: Each line represents a PMCI and the dark red line is the aggregate of all the eight PMCIs. The cumulative percentage is calculated for a month with the total number of individuals registered until the last day of the month from June 2019 and the total population in the PMCI as the denominator. * This is derived by aggregating numerator and denominator from the eight selected PMCIs. The data labels are mentioned for December 2021, December 2022, and December 2023.

**Figure 3 healthcare-11-00575-f003:**
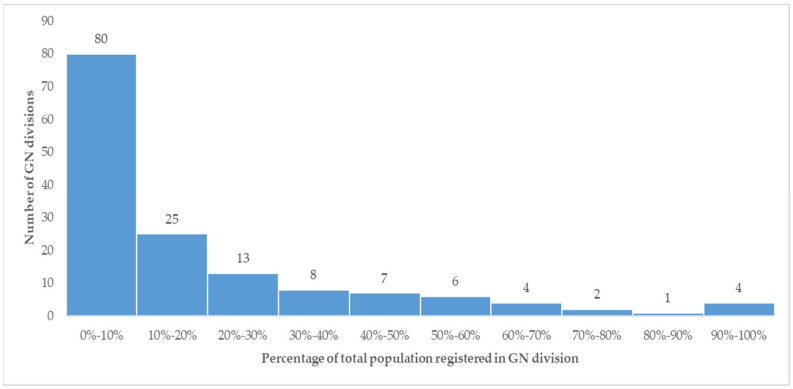
Histogram depicting the number of GN divisions and their percentage of the total population registered in the nine selected PMCIs supported by the PSSP in Sri Lanka, 2021. Note: GN division—Grama Niladhari division (most peripheral administrative unit in Sri Lanka).

**Figure 4 healthcare-11-00575-f004:**
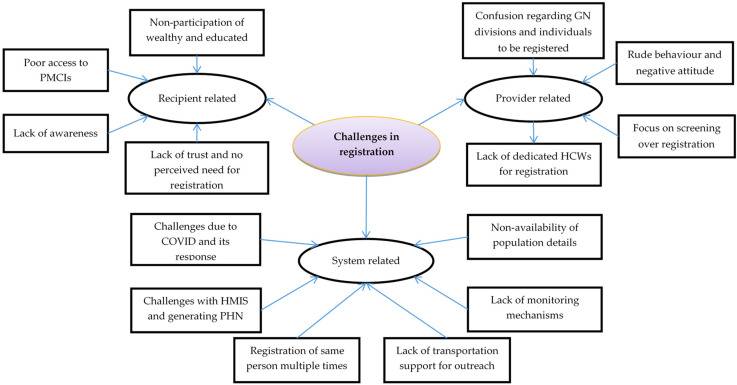
Challenges in registration of the identified population in the PMCIs supported by PSSP in Sri Lanka, 2021. Abbreviation: HCWs: Health Care Workers; PMCI: Primary Medical Care Institution; PSSP: Primary Health Care System Strengthening Project (PSSP); HMIS: Health Management Information System; PHN: Personal Health Number.

## Data Availability

Requests to access these data should be sent to the corresponding author.

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
