# Peer review of "Empanelment of the Population to the Primary Medical Care Institution of Sri Lanka: A Mixed-Methods Study on Outcomes and Challenges"

_healthcare, 2023, doi:10.3390/healthcare11040575_

Round 1
Reviewer 1 Report
1. Projecting estimated coverage using linear trend in Figure 2 is questionable, as the projection period is too far away in the future, and existing data points themselves did not have a clear linear trend.
2. The authors should be more clear about why 35 yrs was selected as the cutoff for the analysis. As younger individuals are naturally less likely to seek care therefore registered. Would a cutoff like 55/65 yrs be more useful when interpreting the results?
3. >100% in Figure 3 is a bit confusing, better incorporate them into 90-100% or further explain
Author Response
Thank you for the detailed review and comments/suggestions. Please find below the response for the comments.
1. Projecting estimated coverage using linear trend in Figure 2 is questionable, as the projection period is too far away in the future, and existing data points themselves did not have a clear linear trend.
Response: Thank you for this comment. We agree on the limitations with the simple linear forward projection using the existing data points. We had to go with the simple linear projection as data did not fit into any other distribution. However, this projection was made to emphasise the deficiency in the rate of registration and its potential implication on the community coverage by the end of the project. Acknowledging the issues with this projection, we have explicitly highlighted the issue with the projection under the limitation section of the discussion. Changes have been made in lines 610 to 613 (track changes with all mark-up mode).
2. The authors should be more clear about why 35 yrs was selected as the cutoff for the analysis. As younger individuals are naturally less likely to seek care therefore registered. Would a cutoff like 55/65 yrs be more useful when interpreting the results?
Response: Thank you for this comment. The registration was supposed to include all the individuals residing in the identified GN divisions. Additionally, PSSP had a mandate for screening individuals aged ≥35 years for NCD risk factors. The registration (empanelment) and screening had to be two separate activities under the project. One of the disbursement-linked indicators was to screen at least 25% of the individuals ≥35 years for NCD risk factors in year-1. Thus, the healthcare providers prioritized registration of individuals ≥35 years and utilized the registration opportunity to screen for NCDs. This has been highlighted well in the qualitative results section of the manuscript. However, we agree with the reviewer’s comment and have explained the reason for choosing the cut-off under the methods section. Changes have been made in lines 255-256 (track changes with all mark-up mode).
3. >100% in Figure 3 is a bit confusing, better incorporate them into 90-100% or further explain
Response: Thank you for this suggestion. We have incorporated >100% under 90%-100%. Changes have been made in Figure 3.
Reviewer 2 Report
Thank you for the opportunity to read the manuscript "Empanelment of the population to the Primary Medical Care Institution of Sri Lanka: A mixed methods study on outcomes and challenges" submitted to Healthcare.
The manuscript offers many interesting first-hand insights into PCMIs in a region that is underrepresented on the map of research. I like the manuscript but I think there is room for improvement in the following points.
1. It is not entirely clear to me who the article is aimed at. If the article is also aimed at an audience outside of Sri Lanka, then a more general overview would certainly be helpful.
2. The authors write that they randomly selected six PMCIs from all PMCIs in rural areas and three PMCIs from all PMCIs in urban areas. What exactly did the selection procedure look like and why is urbanity the most important selection criterion? This is not clear to me.
3. Figure 1 seems a bit overly packed and cluttered to me.
4. It is not clear to me why a projection is needed for the number of registered persons in the PMCIs. The data from 2021 and 2022 should already be available. Or am I missing something?
5. I agree with the authors that the male-to-female ratio (39 vs. 61 percent) is a serious limitation. The authors wrote nothing so far about the implications for interpreting the results. This needs to be addressed.
Author Response
Thank you for the detailed review and comments/suggestion. Please find below the response to the comments/suggestions.
The manuscript offers many interesting first-hand insights into PMCIs in a region that is underrepresented on the map of research. I like the manuscript but I think there is room for improvement in the following points .
1. It is not entirely clear to me who the article is aimed at. If the article is also aimed at an audience outside of Sri Lanka, then a more general overview would certainly be helpful.
Response: Thank you for this comment. The manuscript is primarily aimed at the policy makers in Sri Lanka. However, we feel that the subject of primary healthcare system strengthening is an issue of interest for most of the low- and middle-income countries, and especially the empanelment which is a relatively new concept in primary healthcare. To cater to international readers, under the specific setting of the methods section, we have tried to provide details about the PSSP and the steps involved in the empanelment. We have also provided references to project documents and previous publications while describing the PSSP and the empanelment process. We will be happy to provide any further specific information if required.
2. The authors write that they randomly selected six PMCIs from all PMCIs in rural areas and three PMCIs from all PMCIs in urban areas. What exactly did the selection procedure look like and why is urbanity the most important selection criterion? This is not clear to me.
Response: Thank you for this comment. The reason for choosing 2:1 (rural: urban) was to ensure the study sample is in line with the rural and urban ratio among the general population in Sri Lanka. This would enable us to accommodate for the variation in registration rates in urban and rural areas due to difference in the demographic characteristics. Also, this selection of PMCIs at 2:1 ratio was one of requirements from the funders of this evaluation. Regarding the query on the method of selection, we have mentioned that ‘simple random sampling (computer generated numbers) was used to select PMCIs’ in lines 177-179 (track changes with all mark-up mode).
3. Figure 1 seems a bit overly packed and cluttered to me.
Response: Thank you for this comment. However, as it needs to accommodate trends over time in eight PMCIs and one aggregate measure, we are not able decongest it without losing out on the relevant messages. We hope the reviewer can accept our stance here.
4. It is not clear to me why a projection is needed for the number of registered persons in the PMCIs. The data from 2021 and 2022 should already be available. Or am I missing something?
Response: Thank you for this comment. All the data for study was collected in July 2021. The data on registration was extracted from April 2019 and censored by June 2021. Therefore, we do not have data regarding registration coverage from June 2021 onwards. However, we see that this was not explicitly mentioned in the manuscript. Thus, we have added a sentence on the period of data collection under the methods section. Changes have been made in lines 209-210 (track changes with all mark-up mode).
5. I agree with the authors that the male-to-female ratio (39 vs. 61 percent) is a serious limitation. The authors wrote nothing so far about the implications for interpreting the results. This needs to be addressed.
Response: Thank you for this comment. This was one of the important findings of the study that not many males had registered under the PSSP project (only 39% of the registered people were males). In the discussion, in lines 673 to 682 (track changes with all mark-up mode), we have tried to interpret these results, discussed the implications in detail and also have made certain recommendations to improve the registration of males under PSSP.